# *Acer pseudoplatanus*: A Potential Risk of Poisoning for Several Herbivore Species

**DOI:** 10.3390/toxins14080512

**Published:** 2022-07-26

**Authors:** Benoît Renaud, Caroline-Julia Kruse, Anne-Christine François, Lisa Grund, Carolin Bunert, Lucie Brisson, François Boemer, Gilbert Gault, Barbara Ghislain, Thierry Petitjean, Pascal Gustin, Dominique-Marie Votion

**Affiliations:** 1Department of Functional Sciences, Faculty of Veterinary Medicine, Pharmacology and Toxicology, Fundamental and Applied Research for Animals & Health (FARAH), University of Liège, 4000 Liège, Belgium; acfrancois@uliege.be (A.-C.F.); p.gustin@uliege.be (P.G.); dominique.votion@uliege.be (D.-M.V.); 2Department of Functional Sciences, Physiology and Sport Medicine, Faculty of Veterinary Medicine, Fundamental and Applied Research for Animals & Health (FARAH), University of Liège, 4000 Liège, Belgium; caroline.kruse@uliege.be; 3Der Grüne Zoo Wuppertal, 42117 Wuppertal, Germany; grund@zoo-wuppertal.de; 4Zoo Duisburg gGmbH, 47058 Duisburg, Germany; bunert@zoo-duisburg.de; 5Zoo de Beauval, 41110 St. Aignan, France; lucie.brisson@zoobeauval.com; 6Biochemical Genetics Laboratory, Human Genetics, CHU Sart Tilman, University of Liège, 4000 Liège, Belgium; f.boemer@chuliege.be; 7USC 1233 & Centre National d’Informations Toxicologiques Vétérinaires (CNITV), 69280 Lyon, France; gilbert.gault@vetagro-sup.fr; 8Fourrages Mieux Asbl, 6900 Marche-en-Famenn, Belgium; barbara.ghislain@gmail.com; 9Arsia Asbl, 5590 Ciney, Belgium; thierry.petitjean@arsia.be

**Keywords:** hypoglycin A, methylenecyclopropylacetyl-carnitine, sycamore maple, toxin, poisoning, zoo, gnus, ruminants, equine atypical myopathy

## Abstract

*Acer pseudoplatanus* is a worldwide-distributed tree which contains toxins, among them hypoglycin A (HGA). This toxin is known to be responsible for poisoning in various species, including humans, equids, Père David’s deer and two-humped camels. We hypothesized that any herbivore pasturing with *A. pseudoplatanus* in their vicinity may be at risk for HGA poisoning. To test this hypothesis, we surveyed the HGA exposure from *A. pseudoplatanus* in species not yet described as being at risk. Animals in zoological parks were the major focus, as they are at high probability to be exposed to *A. pseudoplatanus* in enclosures. We also searched for a toxic metabolite of HGA (i.e., methylenecyclopropylacetyl-carnitine; MCPA-carnitine) in blood and an alteration of the acylcarnitines profile in HGA-positive animals to document the potential risk of declaring clinical signs. We describe for the first instance cases of HGA poisoning in *Bovidae*. Two gnus (*Connochaetes taurinus taurinus*) exposed to *A. pseudoplatanus* in their enclosure presented severe clinical signs, serum HGA and MCPA-carnitine and a marked modification of the acylcarnitines profile. In this study, even though all herbivores were exposed to *A. pseudoplatanus*, proximal fermenters species seemed less susceptible to HGA poisoning. Therefore, a ruminal transformation of HGA is hypothesized. Additionally, we suggest a gradual alteration of the fatty acid metabolism in case of HGA poisoning and thus the existence of subclinical cases.

## 1. Introduction

Originally from Central Europe, Caucasus and North of Asia Minor [1], the *Acer pseudoplatanus* (sycamore maple, *Sapindaceae*) is nowadays a worldwide-distributed tree. It has become naturalized in most of Europe and even in other countries and continents, such as North and South America, New Zealand, Australia and India [2]. Its rapid growth, high timber price [3], attractive appearance tolerance to pollution, salt, wind and low summer temperatures [4], make it economically interesting for silviculture and as an ornamental tree. However, despite all its apparent advantages, the seeds [5], the seedlings [6] and, to a lesser extent, the leaves [7] from *A. pseudoplatanus* contain toxins. Among them, hypoglycin A (HGA) is known to be responsible for poisoning in various species, including humans. Hypoglycin A poisoning was first confirmed in humans in the context of the Jamaican vomiting sickness [8,9] and more recently as taking part in outbreaks of encephalopathy [10,11,12,13]. Later on, HGA was the first toxin incriminated in equine atypical myopathy [5,14]. Numerous equids are already described as sensitive to HGA: saddle horses, draft horses, ponies, donkeys, zebras and Przewalski horses [15,16].

To exert its toxicity, HGA must be metabolized in liver cells into methylenecyclopropylacetyl-coenzyme A (MCPA-CoA) [17] which impairs lipid metabolism [18], resulting in an accumulation of acylcarnitines in blood [19,20,21,22]. In veterinary medicine, the detection of both HGA and methylenecyclopropylacetyl conjugated with carnitine (MCPA-carnitine) combined with a modified serum acylcarnitines profile is thought to confirm the diagnosis of HGA poisoning [21,23] but no toxicological standards for these biomarkers of effects have been determined so far. In Europe, *A. pseudoplatanus* poisoning is related to animals on pasture during autumn and spring, which are likely to ingest seeds and seedlings present on the ground in these seasons [5,6,24].

Laboratory research studies have shown that mice, rats, guinea pigs, rabbits, pigeons, dogs, cats and monkeys may be poisoned by HGA [9,25]. Very recently, HGA poisoning has been shown in two new species: Père David’s deer (*Elaphurus davidianus*) [26,27] and two-humped camels (*Camelus ferus bactrianus*) [22]. Thus, for the first time, HGA poisoning was described in foregut fermenters. Despite a variety of clinical signs between species, HGA poisoning can lead to death in all the above-cited ones [9,14,22,25,26]. In grazing equids, atypical myopathy displays the characteristics of an emerging disease [24] while domestic ruminants seem to be spared despite access to grazing land. However, although domestic ruminants may be exposed to a lesser extent (e.g., different pasturing times according to season, different pasture management, etc.) the question of a possible resistance and/or tolerance [28] to HGA poisoning remains unsolved.

In the context of an omnipresence of *A. pseudoplatanus* and its invasive character [2], we hypothesized that any herbivore pasturing with this tree in their vicinity may be at risk for HGA poisoning. To test this hypothesis, we surveyed serum HGA and MCPA-carnitinein animal species not yet described as being at risk. Animals in zoological parks were the major focus as they are mostly kept in enclosures containing/surrounded by trees and are therefore at high risk to be exposed to *A. pseudoplatanus*. We also searched for MCPA-carnitine in blood and an alteration of the acylcarnitines profile in HGA-positive animals in order to contribute to the characterization of the toxicological risks.

## 2. Materials and Methods

### 2.1. Origin of the Samples

All procedures performed in this study were in accordance with the national and international guidelines for animal welfare. Serum samples were selected from available banks of zoological parks which had been taken previously to establish a diagnosis in diseased animals or to prevent atypical myopathy. The remaining samples were collected in one alpaca farm (Aubel, Belgium) and at the farm of the Faculty of veterinary Medicine, University of Liège (Liège, Belgium).

The studied species include herbivorous mammals which are either foregut fermenters or hindgut fermenters. The foregut fermenters were classified according to the continuum of morphophysiological feeding type hypothesized by Hofmann and collaborators [29,30] and now confirmed in vivo [31]. Sample information, species details and effectives are presented in Table 1.

To be included in this study, blood samples provided by zoological parks from their biobank (1) had to belong to grazing animals with *A. pseudoplatanus* in their close vicinity, (2) had to have been sampled during the highest risk periods as previously defined (i.e., in autumn, from October to December and during the following spring from March to May [24] and (3) in years with large outbreaks (i.e., more than 50 cases declared to the Atypical Myopathy Alert Group since 2006 [19]. 

Thirteen blood samples were retrieved from the bank of serum from “Neunkircher Zoo” in Germany, ten from “Der Grüne Zoo Wuppertal” in Germany, eight from “Zoo Duisburg” in Germany and six from “ZooParc de Beauval” in France. The latest were collected in spring 2020 and spring 2021, from gnus (*Connochaetes taurinus taurinus*) with a tentative diagnosis of atypical myopathy.

From the samples taken by veterinary practitioners and sent to AMAG to assess the risk of atypical myopathy and/or to diagnose HGA poisoning, twenty-one blood samples came from an alpaca farm in which alpacas were exposed to *A. pseudoplatanus* seeds and leaves, eight from goats, five from cattle and thirteen from sheep. All animals were sampled during the years and seasons at risk as mentioned above.

### 2.2. Hypoglycin A Assay

Hypoglycin A assay was performed on serum using an aTRAQ^®^ kit for amino acid analysis of physiological fluids as previously described [23,26,32]. Hypoglycin A contained in samples was derivatized using an isotopic tag (mass *m*/*z* 121), while a second labeling reagent (mass *m*/*z* 113) allowed absolute quantification. Derivatized samples were introduced into a TQ5500 tandem mass spectrometer (Sciex) using a Prominence AR HPLC system (Shimadzu). This method presents a lower limit of quantification of 0.090 μmol/L [32].

### 2.3. Methylenecyclopropylacetyl–Carnitine Determination Method

The MCPA-carnitine separation and determination were carried out by ultra-performance liquid chromatography combined with subsequent mass spectrometry (UPLC-MS/MS) as reported by Valberg et al., in 2013. The corresponding limit of detection is approximately 0.001 nmol/L [14].

### 2.4. Acylcarnitines Determination Method

Free carnitine and twenty-one acylcarnitines (C2, C3, C3DC, C4, C5, C5-OH, C5DC, C6, C8, C8:1, C10, C10:1, C10:2, C12, C12:1, C14, C14:1, C16, C16:1, C18 and C18:1 -carnitine) [33] were quantified in serum by tandem mass spectrometry [34]. A methanol solution with labelled internal standards was used to precipitate serum proteins. Supernatants were evaporated under nitrogen stream and derivatized with butanolic-HCl. Samples were then analyzed with a TQ5500 mass spectrometer (Sciex, Framingham, MA, USA).

### 2.5. Statistical Analysis

Partial least squares regression (PLS) was used to compare acylcarnitines’ profiles. This statistical model is particularly fitted as serum acylcarnitines profile correspond to numerous (21) correlated variables. The PLS visual outcome is a two-dimensional projection where each blood sample is represented as one point. Component 1 (*x*-axis) and component 2 (*y*-axis) are weighted combinations of the original variables.

Partial least squares regressions allow for both a discrimination of groups as well as a selection of relevant variables. The identification of relevant acylcarnitines to specify a group of animals was performed using two scores: (1) variable importance in the projection (VIP) and (2) center parameter estimates of the acylcarnitines of main interest with a VIP over 1 and an absolute value of the center parameter estimates over 0.15 [35,36].

Normality was assessed using a Shapiro–Wilk test. Data were log-transformed when necessary to compare different groups of animals (Mann-Whitney test or unpaired *t* test).

Statistical analyses and graphical representations were performed using SAS 9.4M7 (PLS regressions) and Graphpad Prism 7 (Mann Whitney test).

## 3. Results

Based on serum values of HGA and MCPA-carnitine, three groups of animals were characterized: (1) animals negative for HGA and negative for MCPA-carnitine, (2) animals positive for HGA and negative for MCPA-carnitine and (3) animals positive for MCPA-carnitine. The first PLS regression compares serum acylcarnitines profile between the three groups (Figure 1 and Table 2). On the *x*-axis, the component 1 of the regression explains 72.51% of the differences between the three aforementioned groups.

For group 1, the component 1 is minor to 0 for 82% (40/49) of the animals and over 0 for 18% (9/49) of the animals. These nine last cited animals are non-diseased: four gnus (S08, S09, S10, S11), two camels (S59, S61), two cows (S05, S06) and one zebra (S74).

For group 2, the component 1 is minor to 0 for 30% (3/10) of three elephants S65, S68 and S70. Those three elephants are non-diseased animals and presented serum HGA concentrations below than the 20th percentile of the positive HGA observations.

For group 3, the component 1 is minor to 0 for 25% (4/16) of two alpacas (2/22) positive for MCPA-carnitine (S55, S56), the only elephant (1/6) positive for MCPA-carnitine (S69) and one goat (S29). The elephant S69 is a non-diseased animal and presented the lowest MCPA-carnitine serum value of the positive MCPA-carnitine observations.

The inclusion criteria did not incorporate a specific clinical status. Most of the animals were healthy in appearance (i.e., no clinical signs were observed during general observational examination). Of the 75 blood samples from 70 animals, two gnus (S12, S13) were diseased at the time of sample collection. Information about the diseased animals, clinical signs and complementary examination results are gathered in Table 3. The second PLS regression compares serum acylcarnitines profile between two groups: (i) animals which presented clinical signs at the time of blood sampling, (ii) animals without clinical sign at that time (i.e., animals healthy in appearance; Figure 2).

On the *x*-axis, the component 1 of the regression explains 73% of the difference between clinically affected animals and non-diseased ones. According to this component: the acylcarnitines profile of the two diseased gnus (S12 and S13) deeply differ from animals healthy in appearance. The acylcarnitines profile of two of the five camels (S59 and S61) are distinct from the apparently healthy animals without being as extreme as the diseased gnus.

## 4. Discussion

The clinical poisoning associated with the ingestion of HGA from *A. pseudoplatanus* has been described in a few species. Initially, this environmental poisoning was reported in equids (i.e., horses, ponies, donkeys, Przewalski horses and zebras) as equine atypical myopathy [16,37]. The condition was first described in 1985 [38] but was only linked to the *A. pseudoplatanus* in 2014 [39]. More recently, clinical signs and death of camels and Père David’s deer were connected to HGA poisoning through ingestion of *A. pseudoplatanus* seedlings [22,26].

We report here the presence of HGA and/or MCPA-carnitine in several species besides equids. The absence of HGA and MCPA-carnitine in serum from both donkeys and zebras suggests that the toxic material was not available in sufficient quantities in their enclosures when they were sampled.

In one sheep, one goat and two alpacas, MCPA-carnitine was detected while no HGA was present in the serum. This may be due to the involved toxicokinetic. It is plausible that HGA had already been eliminated from their blood while MCPA-carnitine was still present. This hypothesis is sustained by previous descriptions in the literature. Indeed, in one horse removed from the toxic environment, it was shown that MCPA-carnitine increased in the serum with time while HGA decreased [40]. Moreover, similar results were obtained in human poisoning with ackee fruits: after HGA maximal serum concentration was attained, its concentration decreased while MCPA-carnitine increased [41].

### 4.1. Subclinical Status

This study underlines the existence of yet undescribed subclinical HGA poisoning in species other than equids. Apparently healthy animals that tested positive for HGA presented a global increase of the serum acylcarnitines profile (component 1 comparison in Figure 1 and Table 4). This observation should be investigated further in equids, which represent most cases of *A. pseudoplatanus* poisoning. The apparently healthy animals with measurable serum HGA and/or MCPA-carnitine can be assimilated to healthy co-grazing horses in literature [42]. Co-grazing animals present no clinical signs but are exposed to *A. pseudoplatanus* as they pasture with horses suffering from atypical myopathy [6,42,43,44]. Apparently healthy animals with no detected HGA and MCPA-carnitine in serum can be assimilated to control horses [6,44]: they are healthy and not exposed to the toxin. In animals that were healthy in appearance but tested positive for MCPA-carnitine, the levels of free carnitine and acylcarnitines profile were higher compared to “controls” (i.e., non-exposed animals; Mann–Whitney test, *p* = 0.0019, Table 4), but lower compared to diseased animals (Mann–Whitney test, *p* = 0.0062).

Regarding horses, the literature tends to describe *A. pseudoplatanus* poisoning like an on/off disease correlated with a multiple acyl-CoA dehydrogenases phenotype (MADD; [33,45]). It appears in this study that HGA poisoning might not be binary and probably results from a gradual fatty acid metabolism alteration.

### 4.2. Rumenal Hypoglycin A Metabolization

We describe for the first time HGA poisoning in *Bovidae*. Two gnus (*Connochaetes taurinus taurinus*) exposed to *A. pseudoplatanus* in their enclosure presented severe clinical signs in the spring of 2021. In blood samples collected at that time, HGA and MCPA-carnitine were detected. This observation confirmed (i) exposition to HGA, (ii) HGA absorption and (iii) HGA metabolization into its active metabolite. Furthermore, in these two gnus (S12 and S13), a severe modification of the acylcarnitines profile was observed (Figure 2). The global increase of acylcarnitines has such a large amplitude that it masks the interspecies variation. This phenomenon reflects an alteration of the lipid catabolism, which is associated with the modified blood parameters, similar clinical signs as in equine atypical myopathy. Added to the exposition to *A. pseudoplatanus* during the risky season, there is a high probability that the two gnus were poisoned by toxins of this tree. Moreover, this poisoning seems to follow a similar pathophysiological pathway to the one described in atypical myopathy with equids.

Like equids, some foregut fermenters (Père David’s deer, gnus) are now proven susceptible to this poisoning. This raises the question of the difference of sensitivity between species under a similar toxic pressure. Equids are, besides foregut fermenters, the species most subjected to *A. pseudoplatanus* poisoning.

Hypoglycin A is a small non-proteogenic amino acid. It is highly hydrophilic and soluble. Once ingested, HGA absorption takes place in the proximal small intestine [46]. The presence of a fermentation compartment proximal to the small intestine might be protective. Indeed, the toxic molecule may be transformed by rumen microbiota even more when the rumen retention time is long.

In equids, which have a fermenter compartment (i.e., caecum) distal to the HGA absorption area (i.e., small intestine), most of the ingested HGA is available for absorption. Serum HGA has been detected in mixed feeders (goats, sheep) [47]. In mixed feeders, HGA absorption could be explained by short rumen retention time and large salivary production coupled with a highly functional ventral groove [30,48]. Following initial chewing, the soluble amino acids, including HGA, bypass ruminal fermentation which leads HGA to be absorbed within the proximal small intestine.

Hypoglycin A ruminal transformation (and possibly degradation or metabolization) may explain why HGA was not observed in serum of cattle. The energetic strategy of these large grazers takes better advantages on fermentations with a big relative rumen size and a long rumen retention time. The rumen chyme is less liquid and the ventral groove less developed which both limit the rumen bypass [30,48]. It has to be noted that HGA has been detected in cow’s milk [49]. In those cows, some HGA must have gotten through the rumen, which leads to the hypothesis that, under a high toxic pressure, even these grazers might be at risk of HGA poisoning.

The hypothesis that retention time of soluble molecules plays a role in the risk of HGA poisoning concurs with the observations in camelids. Indeed, camelids have previously been described as susceptible for clinical HGA poisoning [22], and despite having a relative rumen size and rumen microbiota close to the cattle [50,51], the camelids rumen retention time for liquids is short. The retention time is even shorter than in small ruminants [52,53,54] which makes ingested HGA likely to pass to the small intestine and to be absorbed. In a recent article, González-Medina and collaborators (2021) observed that HGA is not affected by a 2 h incubation time with ovine rumen fluid. This observation does not exclude a HGA rumen transformation as the ovine rumen retention time is 7–35 h for solutes and 10–50 h for dry matter [55,56].

Elephants, who are distal fermenters, might have, as equids, a high probability of HGA intestinal absorption when ingested. Four of the six sampled elephants were positive for HGA or MCPA-carnitine in their blood. Despite living on solid flooring with no possibility of directly eating *A. pseudoplatanus* seedlings, the elephants were still exposed to HGA from *A. pseudoplatanus* samaras, leaves and possibly contaminated hay or water. In these elephants, the concentrations of HGA were amongst the lowest of the observed and were not associated with a global increase of the serum acylcarnitines profile. This could be related to low quantities of serum HGA/MCPA-carnitine or to a different HGA impact on elephants’ fatty acid catabolism.

If the morphophysiology of the gut is most probably linked to the risk of HGA poisoning, the feed intake might also play a part. This is especially relevant in animals kept in zoos, where the feeding supply and program can be challenging and divergent from natural ones. It must be mentioned that even though most atypical myopathy cases in equids are reported during autumn, the camels’ and the gnus’ occurrences took place during spring (i.e., resulting from seedlings ingestion). Our hypothesis at this point drifts from the discussion above. Spring feed might be waterier, especially when the HGA ingestion is linked to *A. pseudoplatanus* seedlings. This might be linked to the liquid chyme, which washes HGA through the rumen, decreasing the time the toxic molecule stays in the rumen. This way, most of the ingested HGA would avoid fermenter transformation and would favor its absorption.

As *A. pseudoplatanus* are largely distributed and since most of the herbivore species kept in zoos might be susceptible to HGA poisoning, measures should be taken to reduce exposure to the *A. pseudoplatanus* samaras and seedlings. Some of the protective measures described for sport or leisure equids [24] seem practically not implementable in zoos. Decreasing time on pasture, rotating meadows or seasonally moving to a low-risk enclosure would be unachievable in most cases. It appears necessary to assess the risk of exposition to *A. pseudoplatanus* before settling animals in a new enclosure. When the enclosures are exposed, the toxic pressure to the animals can be reduced by collecting the samaras or seedlings. Mowed vegetal material deserve a careful attention as it still represents a risk if left on the ground [57]. Other vegetal species associated with HGA poisoning should undergo the same preventative measures than *A. pseudoplatanus* [9]. This applies in particular to *Acer negundo* and *Acer palmatum* spp. commonly found as ornamental trees in zoological parks [58].

## 5. Conclusions

*Acer pseudoplatanus* is proven as responsible for clinical poisoning in camels, Père David’s deer and now in gnus, with a similar pathophysiological pathway to the one described in equine atypical myopathy.

In this study, even though all examined herbivores were exposed to *A. pseudoplatanus*, proximal fermenters species seem to be less susceptible to HGA poisoning. Their gut morphophysiology may act as a protection as toxins may be transformed in the rumen. Indeed, the degree of protection against *A. pseudoplatanus* may be directly linked to the rumen retention time of soluble molecules.

Besides, we described a global increase of the serum acylcarnitines profile in apparently healthy animals which presented HGA or MCPA-carnitine in their blood. This new reported observation suggests a gradual fatty acid metabolism alteration in case of HGA poisoning and thus the existence of subclinical cases.

## Figures and Tables

**Figure 1 toxins-14-00512-f001:**
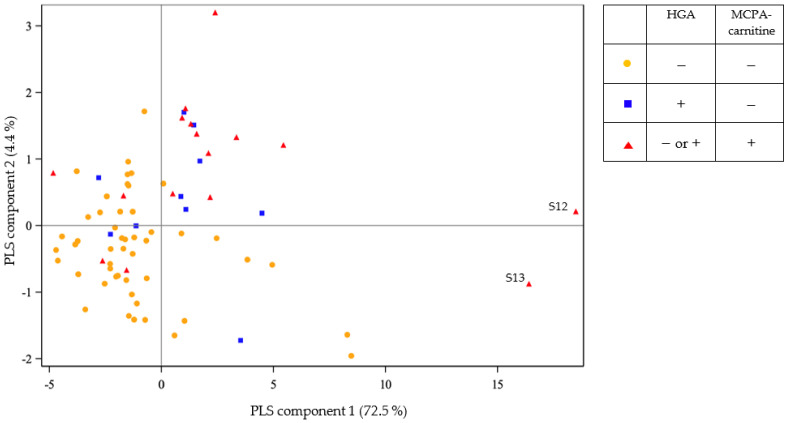
Partial least squares projection of the acylcarnitines profile comparing animals depending on serum hypoglycin A (HGA) and methylenecyclopropylacetyl conjugated with carnitine (MCPA-carnitine) status.

**Figure 2 toxins-14-00512-f002:**
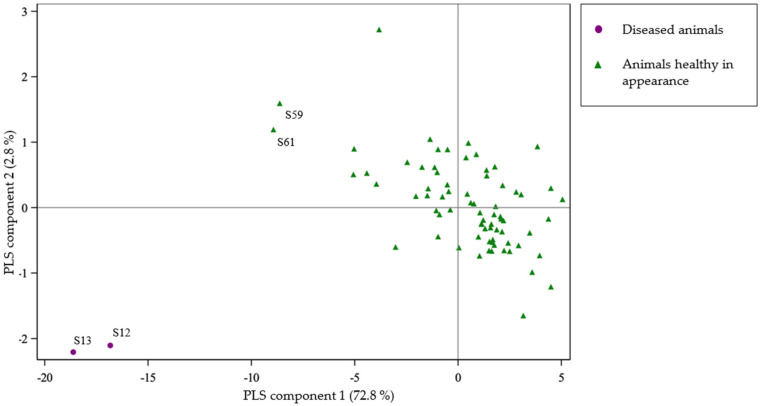
Partial least squares projection of the acylcarnitines profile comparing diseased animals versus apparently healthy ones.

**Table 1 toxins-14-00512-t001:** Individuals included in this study.

Species	Effectives	Sample Number	Sampling Date (dd/mm/yyyy)	Origin	HGA (μmol/L)	MCPA- Carnitine (nmol/L)
Foregut fermenters—Ruminants—Grazers						
Cattle (*Bos taurus*)	7	S01 S02 S03 S04 S05 S06 S07	10/12/2018 10/12/2018 10/12/2018 10/12/2018 10/12/2018 21/05/2021 21/05/2021	Uliège farm Uliège farm Uliège farm Uliège farm Uliège farm Neunkirchen Neunkirchen	- - - - - - -	- - - - - - -
Gnu (*Connochaetes taurinus taurinus*)	6	S08 S09 S10 S11 S12 * S13 *	14/04/2020 04/03/2021 16/03/2021 01/04/2021 29/04/2021 29/04/2021	Beauval Beauval Beauval Beauval Beauval Beauval	- - - 0.16 0.28 0.67	- - - - 0.30 4.67
Blackbuck (*Antilope cervicapra*)	1	S14	22/04/2016	Neunkirchen	-	-
Foregut fermenters—Ruminants—Intermediate feeders						
Sheep (*Ovis aries*)	15	S15 S16 S17 S18 S19 S20 S21 S22 S23 S24 S25 S26 S27 S28	01/12/2016 17/05/2017 10/12/2018 10/12/2018 10/12/2018 10/12/2018 10/12/2018 10/12/2018 10/12/2018 10/12/2018 10/12/2018 10/12/2018 10/12/2018 10/12/2018	Neunkirchen Neunkirchen Uliège farm Uliège farm Uliège farm Uliège farm Uliège farm Uliège farm Uliège farm Uliège farm Uliège farm Uliège farm Uliège farm Uliège farm	- - - - 0.20 0.76 0.84 0.99 1.01 1.11 1.13 1.38 2.07 2.77	- - - 3.18 - - - 3.78 13.20 3.64 - 5.26 3.75 8.21
Goat (*Capra hircus*)	13	S29 S30 S31 S32 S33	29/10/2016 08/11/2018 27/04/2020 27/04/2020 27/04/2020	Wuppertal Wuppertal Duisburg Duisburg Duisburg	- - 0.55 0.82 1.25	0.14 - - 1.15 0.79
Timor deer (*Rusa timorensis*)	1	S34	01/10/2015	Neunkirchen	-	-
Foregut fermenters—Ruminants—Camelids						
Alpaca (*Vicugna pacos*)	22	S35 S36–S54 S55 S56	29/05/2015 12/10/2018 12/10/2018 12/10/2018	Neunkirchen Alpaca farm Alpaca farm Alpaca farm	- - - -	- - 0.32 1.32
Camel (*Camelus bactrianus*)	5	S57 S58 S59 S60 S61	17/04/2006 24/09/2009 25/04/2012 02/04/2014 14/12/2018	Duisburg Duisburg Duisburg Duisburg Duisburg	0.38 - - - -	- - - - -
Lama (*Lama glama glama*)	2	S62 S63	01/12/2016 17/05/2017	Neunkirchen Neunkirchen	- -	- -
Vicuna (*Vicugna vicugna*)	1	S64	27/04/2020	Wuppertal	0.16	1.71
Hindgut fermenters						
Elephant (*Loxodonta africa*)	6	S65 S66 S67 S68 S69 S70	06/11/2014 17/11/2016 17/11/2016 15/12/2016 28/11/2017 08/11/2018	Wuppertal Neunkirchen Neunkirchen Neunkirchen Wuppertal Wuppertal	0.16 - - 0.21 - 0.12	- - - - 0.18 -
Donkey (*Equus asinus*)	3	S71 S72 S73	16/12/2016 20/10/2017 12/11/2018	Neunkirchen Wuppertal Wuppertal	- - -	- - -
Zebra (*Equus quagga boehmi*)	2	S74 S75	30/10/1999 30/10/2007	Wuppertal Wuppertal	- -	- -

* Diseased animals. The “-” stands for below the limit of quantification.

**Table 2 toxins-14-00512-t002:** Coefficients in selecting variables of main interest-PLS 1. Component 1 (Mean ± SDM) and serum concentration (μmol/L) for the 3 groups of animals-PLS 1.

	VIP	Center Parameter Estimates
C8:1-Carnitine	2.42	0.63
C10:2-Carnitine	1.25	−0.56
C14:1-Carnitine	1.05	0.26
C12:1-Carnitine	1.04	−0.22
C16:1-Carnitine	1.01	−0.16

Only the 5 acylcarnitines of main interest are presented in this table.

**Table 3 toxins-14-00512-t003:** Diseased animals’ information.

Species	Sample Number	Clinical Signs	Complementary Exams	Outcome
Gnu (*Connochaetes* *taurinus taurinus*)	S12	Depression Tremors Decubitus	CK = 15,000 * IU/L LDH = 20,800 * IU/L AST = 10,688 * IU/L ALT = 1621 * IU/L	Good evolution: clinical signs disappeared within 3 days after removing access to toxin plants
	S13	Depression Tremors Decubitus	CK = 16,800 * IU/L LDH = 10,700 * IU/L AST = 1000 * IU/L ALT = 375 * IU/L	Good evolution: clinical signs disappeared within 3 days after removing access to toxin plants

* Over the reference range. CK, serum activities of creatine kinase with the reference range of 58–502 IU/L. LDH, serum total lactate dehydrogenase activity with the reference range of 15–1591 IU/L. AST, serum aspartate aminotransferase activity with the reference range of 59–230 IU/L. ALT, serum alanine transaminase activity with the reference range of 18–64 IU/L.

**Table 4 toxins-14-00512-t004:** Component 1 (Mean ± SDM) and serum concentration (μmol/L) for the 3 groups of animals-PLS 1.

	HGA − MCPA − (*n* = 49)	HGA + MCPA − (*n* = 10)	HGA − or + MCPA + (*n* = 16)
Component 1	−1.087 ± 0.390 ^a b^	0.818 ± 0.735 ^a^	2.818 ± 1.557 ^b^
C8:1-Carnitine	0.018 ± 0.001 ^c d^	0.030 ± 0.005 ^c^	0.068 ± 0.018 ^d^
C10:2-Carnitine	0.007 ± 0.001 ^e^	0.008 ± 0.001	0.014 ± 0.005 ^e^
C14:1-Carnitine	0.020 ± 0.006 ^f^	0.032 ± 0.011	0.219 ± 0.133 ^f^
C12:1-Carnitine	0.038 ± 0.013	0.022 ± 0.008	0.049 ± 0.025
C16:1-Carnitine	0.045 ± 0.015 ^g^	0.033 ± 0.009	0.649 ± 0.424 ^g^

Only the 5 acylcarnitines of main interest are presented in this table. SDM stands for standard deviation mean. ^a^ Mann–Whitney test, *p* = 0.0133. ^b^ Mann–Whitney test, *p* = 0.0019. ^c^ Unpaired *t* test on log transformed data, *p* = 0.0038. ^d^ Unpaired *t* test on log transformed data, *p* < 0.0001. ^e^ Unpaired *t* test on log transformed data, *p* = 0.0235. ^f^ Unpaired *t* test on log transformed data, *p* = 0.0395. ^g^ Unpaired *t* test on log transformed data, *p* = 0.0142.

## Data Availability

The data presented in this study are available in this article.

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
