# Peer review of "Acer pseudoplatanus: A Potential Risk of Poisoning for Several Herbivore Species"

_toxins, 2022, doi:10.3390/toxins14080512_

Round 1
Reviewer 1 Report
The article ‘Acer pseudoplatanus: a potential risk of poisoning for several 2 herbivore species;’ provides some interesting observations of potential toxicity of HGA from Acer pseudoplatanus in a natural setting.
The paper is well written and is presented in a manner that is easy for the reader to follow. Although longitudinal measurements for variations of HGA and HGA metabolites - following changes in exposure to Acer pseudoplatanus products would have provided support for the authors conclusions - this study is within a natural setting and these follow-up analysis can be difficult to complete. Moreover, one of the strengths of this manuscript, is the information collected is within a natural (non- controlled) environment.
Prior to publication however, and in this reviewer’s opinion, several comment should be addressed.
Line 87. The authors indicated that ‘exposure’ to Acer pseudoplatanus was criteria for evaluating samples collected from various animal species. If exposure refers to animals being in the vicinity of Acer pseudoplatanus, that does not conclusively demonstrate that animals ingested Acer pseudoplatanus products and can not unequivocally confirmanimal exposure. As such the wording could be adjusted to reflect that difference.
Line 284 and Table 3. Related to the previous comment, its appears undetermined if the animals ingested Acer pseudoplatanus products or the animals were only in the vicinity of the trees. In addition, it is only suggestive and non- conclusive that resolution of the clinical sign or disease were associated with a reduction in HGA and HGA metabolites as subsequent measurements serum levels of HGA and HGA metabolites following relocation of Gnus were not completed. Therefore, this wording should be also be adjusted to reflect this difference
Author Response
Dear reviewer, thanks you for the time you spent reviewing this manuscript. Please find my answers in the attached document. Best regards.

Reviewer 2 Report
In the manuscript entitled "Acer pseudoplatanus: a potential risk of poisoning for several herbivore species", the authors present hypoglycin and hypoglycin metabolites exposure in several herbivore species, including domestic ruminants and wild species held in zoos. They found that most of the tested species could be at risk of hypoglycin poisoning and described the clinical signs displayed by gnus.
The manuscript is very readable but still should undergo an english check. Data are overall clearly presented but some clarifications are required in tables and figures. Statistical analysis should be explained with more details. Some claims should be rewritten to better highlight the scientific evidences backing up those statements. Other minor points should be addressed as detailled hereafter.
L.40: "All equids appear sensitive to HGA since zebras have also succumbed from HGA poisoning". The statement, although possibly true, cannot be made solely based on the death of zebras.
L.57-60: "Despite a variety of clinical signs between species, HGA poisoning can lead to death in every single one [9,14,16,17,19]. In grazing equids, atypical myopathy displays the characteristics of an emerging disease [26] while domestic ruminants seem to be spared despite access to grazing land." Please precise which species are concerned in the first sentence, as later you indicate ruminants being protected (hence not dying).
L.79-80: No samples originating from the experimental station could be found in the results. Please delete mentions to these samples, or add them in table 1.
L.127-140: Statistical analysis should be rewritten. Several sentences are not understandable in their current form ("Partial Least Squares regression (PLS) was used to compare acylcarnitines’ profiles as acylcarnitines are correlated variables."). Also please provide more details for the PLS. In addition, please specify which software you used.
L.149: what is "latest" referring to, as only group 1 and component 1 are mentionned ? Please rephrase.
L. 159-160: The lack of clinical examination should not be considered to classify an animal as healthy or not. Please rephrase.
Table 1:Alpacas could be grouped or clearly spelled out whether the samples were from AMAG or Duisburg
Also, I suggest to replace "limit of quatification" by "detection threshold"
Table2: not clear at all, as a) and b) could be either coefficient and Mean+- SDM or be referring to superscripts in footnote. Please clarify. Also, Component 1 carnitine should be written in a smaller font to fit in one single line.
Figure 1 and 2: Components should be clearly written in abscissa and ordinate, not only in caption
Table 3: "after removal access to toxin plants". Please rephrase in proper english.
L.244-247: avoid the repeated use of "indeed", twice here in four lines.
L.253-271: I do not agree with the paragraph dealing with a "subclinial status". Poisoning is a matter of dosis. One can suffer from water poisoning, however well hydrated people cannot be described as "subclinically water poisoned". Poisoning could be considered to manifest clinically by definition : "Poisoning is injury or death due to swallowing, inhaling, touching or injecting various drugs, chemicals, venoms or gases." (mayo clinic). One way to circumvent this would be to talk about "accumulation" instead of poisoning.
L.274: Bovidae
L. 284: "conclude that these two gnus were poisoned by toxins of this tree." Data presented allow to strongly suggest a HGA poisoning in these gnu at best, by no way to conclude. In that case Koch's postulates (or any modified version for chemicals and poisons) are far from being satisfied.
L.314: remove "big"
Author Response

(The authors gave the same response as above.)

Reviewer 3 Report
This article presents relevant information about poisoning by sycamore maple and hypoglycin A in animals. The study methodology is adequate and the work was properly conducted. However, I have a few observations to make.
Use poisoning instead of intoxication to avoid misunderstandings.
L.59-63: Domestic ruminants are not mentioned in the first paragraph.
L.70: Italicize the scientific name.
L.239-242: What is the clearance time of HGA and MCPA-carnitine in serum? A fast clearance would lead to a short detection period after consumption of leaves.
L.243-251: Toxicokinetic data would help explain why an animal is positive for MCPA-carnitine but negative for HGA.
L.253-271: Why did you consider some animals to have developed a subclinical poisoning? Changes in serum biochemistry? HGA-positive animals may be tolerant to/resistant to this compound.
L.307-308: The absorbed HGA probably resulted in serum levels below the method´s detection limit, but the concentrated residues in milk enabled detection possible.
L.311-320: As an alternative hypothesis for retention time, the differences in rumen microbiota between cattle and camelids (even though sligth differences) could account for the differences in the toxin biotransformation.
Author Response

(The authors gave the same response as above.)

Reviewer 4 Report
In this article, the authors report the detection of hypoglycin A (HGA) and two metabolites in serum samples obtained from zoos.
After reading through the article my initial reaction was to wonder why the authors did not focus the article on the poisoning of the two gnus. The poisonings were the most novel aspect of this work.
Specific Comments:
This work would be much improved if the authors included the chemical detection and analysis of plant samples from the zoos where HGA and its metabolites were detected in serum samples. Concentrations of toxins in plant material can vary significantly form location to location. It would be an important addition to this work and useful to associate specific plant populations with subclinical poisonings. Were the subclinical poisons due the lesser concentrations of HGA in the plant? Challenging to believe the conclusions about different types of fermenters without knowledge of the exposure or even an estimate of dose. Without HGA exposure information the conclusions presented in this article are problematic.
I found the statistical analyses somewhat confusing. Multiple tests were used but not exactly described. For example, the materials and methods section describes two normality tests but later in the article it was unclear exactly which one was used. I was also left to wonder about the statistical tests. There were only two gnus that showed clinical signs and statistical tests require at least three to be valid. Makes it difficult to draw valid conclusions with such a small dataset.
I didn't see the point of figure 1, found figure 2 useful because there was a clear difference between the diseased and healthy animals.
Author Response

(The authors gave the same response as above.)

Round 2
Reviewer 4 Report
Suitable for publication.